# Numerical Analyses of Wave Generation and Vortex Formation under the Action of Viscous Fluid Flows over a Depression

**Chih-Hua Chang** [1,2]

[1]  Department of Information Management, Ling-Tung University, Taichung 408, Taiwan;
changbox@teamail.ltu.edu.tw
[2]  Natural Science Division in General Education Center, Ling-Tung University, Taichung 408, Taiwan

**Abstract:** Transient free-surface deformations and evolving vortices due to the passage of flows over a submerged cavity are simulated. A two-dimensional stream function–vorticity formulation with a free-surface model is employed. Model results are validated against the limiting case of pure lid-driven cavity flow with comparisons of the vortical flow pattern and velocity profiles. The verification of the free-surface computations are also carried out by comparing results with published potential flow solutions for cases of flows over a depressed bottom topography. The agreements are generally good. Investigations are extended to other viscous flow conditions, where the cavity is set to have the normalized dimension of one by one when scaled by the still water depth. The free-surface elevations and streamline patterns for cases with Froude numbers ranging from 0.5 to 1.1 and different Reynolds numbers ($Re = 5000$ and $500$) are calculated. At the condition of near-critical flow ($Fr \approx 1.0$), the phenomenon of upstream advancing solitons is produced. Viscous effects on the free-surface profile reveal that at a lower value of Re (e.g., $Re = 500$) larger advancing solitary waves are generated. Vortical flow patterns in the cavity are examined for the cases with $Fr = 1.0$ and various values of $Re$. When $Re = 5000$, the vortex pattern includes a primary and a weak, but dominated secondary vortices at the time reaching a nearly quasi-steady motion. For the case of lower $Re$ (e.g., $Re = 500$), a steady-state vortex pattern can be established with a clockwise primary vortex mostly occupied inside the cavity.

**Keywords:** solitons; free-surface flow; depressed bottom; viscous fluid flow; vortical flow

## 1. Introduction

In 1834, John Scott Russell discovered what he called a form of a large solitary elevation or later a solitary wave that was emerged in front of a tow-boat after its sudden stop. Since then, describing the solitary wave elevations produced in shallow water either by flow passing through a disturbance or by a moving object has become a fascinating and important research topic. Most disturbances considered in the past were either a surface pressure or a bottom-placed convex object and the use of a bottom cavity, as a submerged object in general is limited. The viscous effect is also ignored. This paper presents the novelty of a numerical study to explore the transient phenomena of wave generation and vortex evolution resulting from a uniform flow passing over a bottom cavity in shallow water.

A cavity-like region is usually observed in a hydraulic channel or a dredged waterway for navigation use. In nature, various materials—such as toxic chemicals, nutrients, planktons, sludge, or sediments—may be deposited and trapped in concaved bottom trenches or in depressed zones of water-covered terrains. Numerical simulation of vortex motions in a cavity can provide critical information to understand the transport mechanism of materials in it. On the other hand, modeling

cavity flow is also a classical problem for the study of vortex phenomena in fluid mechanics. A square or rectangular cavity has been commonly used as a test domain for theoretical, numerical, and experimental studies of formed vortices. With the continuous advancement of the research tools, either numerically or experimentally, the results generated from the setting of this simple geometry can demonstrate several important concepts of flow motion. For example, Ryzhov and Koshel [1] and Ryzhov et al. [2] discussed the motion change of point vortices due to boundary current in different circular cavity apertures using the Kirchhoff-Routh stream-function method. It is a simple geometric assumption, but may provide valuable qualitative insight into feasible vortex motion near curved coastlines.

In general, four research topics related to the cavity flow can be defined by the following. Type (1): The studies involve a pure lid-driven cavity flow, i.e., typical flow patterns generated by a lid moving at a constant speed (see Shankar and Deshpande [3]). Type (2): The studies focus on the flows and vortices induced by a uniform flow passing over an open cavity. For example, Chang et al. [4] investigated the differences of the evolved vortices between the conditions of laminar and turbulent flows in a three-dimensional cavity. Zhang and Rona [5] examined the pressure distribution that follows a pressure wave passing through a cavity and Fang et al. [6,7] modeled the process of contaminant removal from a cavity. Type (3): The studies considered cavities of various shapes or rectangular cavities with different aspect ratios, such as the visualization photo of triangular cavity flow shown in the book *An Album of Fluid Motion* by Van Dyke [8]. Later, a similar problem was investigated numerically by Erturk and Gokcol [9]. In addition, Chang and Cheng [10] studied lid-driven air flow within an arc-shape cavity. Yin and Kumar [11] explored the induced flow patterns in a cavity with a flexible boundary and with various aspect ratios. Type (4): The cases include the obstacles in a cavity, which form multiple connected domains. For example, Khanafer and Aithal [12], using a finite element formulation, studied the mixed convective flow and heat-transfer characteristics in a lid-driven cavity that contains a circular cylinder.

The present study investigates the evolving flow patterns and generated vortices as a uniform flow with a free surface passing over a submerged cavity, which is different from the usual lid-driven cavity flow as the so-called rigid lid is replaced by a layer of water that moves with a specified speed on top of the cavity. The formed vortices, instead of confining within the cavity, may expand out of the cavity to interact with the external flow outside of the cavity. The movable free surface may also interact with the flow inside the cavity.

Water waves are naturally generated by external forces, such as wind blowing on the water surface, a landslide moving into water (Grilli et al. [13]), a fish swimming in water (Adkins and Yan [14]), moving vessels (Kara et al. [15]), or the movement of a submerged body (Chang and Wang [16]). In the past, researchers explored the wave patterns of flow passing through a submerged object based on the assumption of steady flow at supercritical or subcritical flows. For example, Hanna et al. [17] applied the Schwartz–Christoffel transformation technique and a series truncation based computational procedure to solve the problem of a steady supercritical flow over a trapezoidal obstacle. Furthermore, a steady turbulent flow model was used by Tzabiras [18] to study the super- and subcritical flows over a hump. In viewing a relative motion, interesting phenomena of generation of solitary waves by an external forcing moving at a transcritical speed has also been studied. The features of the generation of upstream-propagating solitary waves were first investigated numerically by Wu and Wu [19] with solutions that were obtained by solving the generalized Boussinesq equations under the conditions of a surface pressure disturbance moving at a speed close to the transcritical regime. Other external forces in similar studies found the literature included a seabed protrusion or a submerged translating obstacle. The hump-like topography is considered to function as a positive external forcing. On the contrary, a depressed topography is referred to as a negative forcing function (Zhang and Chwang, [20]). The features of wave systems generated by a positive or a negative moving forcing are quite different [20]. To the authors' knowledge, the numerical simulation considering a negative forcing function is more challenging than the cases with a positive one. In other related studies, Grimshaw and

Smyth [21], Wu [22], and Camassa and Wu [23] have applied either forced KdV equations or generalized Boussinesq equations to model solitary waves generated by a moving negative forcing. Grimshaw et al. (2009) [24] investigated flow over a hole using fKdV equations. Their study was inspired by the idea that a semi-infinite positive step generates only an upstream-propagating undular bore (Grimshaw et al. (2007) [25]), and a negative step generates only a downstream-propagating undular bore. Zhang and Chwang [20] also developed a numerical model based on the Euler equations to analyze the roles of the positive and negative forcing functions played on the generated waves. In recent times, Xu and Meng [26] investigated the solitary waves generated by a submerged two-dimensional foil with an angle moving in shallow water at subcritical, super-critical and hypercritical speeds.

In the past, the viscous effect has often been ignored in water wave studies, including those described on wave generation. It appears that the first numerical calculation considering the fluid viscosity for solving the soliton radiation problem was carried out by Chang and Tang [27]. Only a smooth bottom hump was considered, and hence, no vortex wake was found in their simulated results. Zhang and Chwang [28] numerically solved the Navier–Stokes equations to study the transcritical flow passing through a submerged semi-elliptical cylinder. They showed that the vorticities were transported and diffused around the body, but no obvious vortex wake appeared behind the streamlined body. Later, Lo and Yang [29], adopted the vorticity-velocity formulations in calculating the flow passing over a blunt body and indicated that the vortices were clearly observed on the leeside of the body. Employed the stream-function–vorticity formulations with a free surface (SVFS) model—extended from a model developed by Tang and Chang [30]; Chang et al. [31] investigated the vortex patterns generated by a near-critical flow passing over a small square hump at the bottom.

The present study applies the further improved SVFS model with a finer and nonuniform grid system to model the flow motions of the viscous fluid encountering a cavity forcing. To validate the numerical model, the patterns of vortices in the cavity are compared with the results from the limiting cases of pure cavity flow studies. For the free-surface profiles, the present solutions are compared with those from the study of soliton radiation produced by a negative forcing function in potential flow. After the confirmation of model validation, the SVFS is applied to simulate the present study of a viscous uniform flow over a cavity. The cases studied for flows over a cavity include the combinations of two different Reynolds numbers ($Re$ = 500 and 5000) and various Froude numbers ($Fr$ = 0.5 to 1.1). The definitions of $Re$ and $Fr$ are stated in the next section. Free-surface deformations and the evolution of vortices are the focuses of this study and analysis.

## 2. Mathematical Formulations

The physical problem is sketched in Figure 1, where the free-surface deformations and vortex evolution during the process of a uniform flow interacting with a cavity are investigated. In the study domain, a two-dimensional (2-D) unsteady viscous flow is assumed. The governing equations, along with the initial and boundary conditions, are described below.

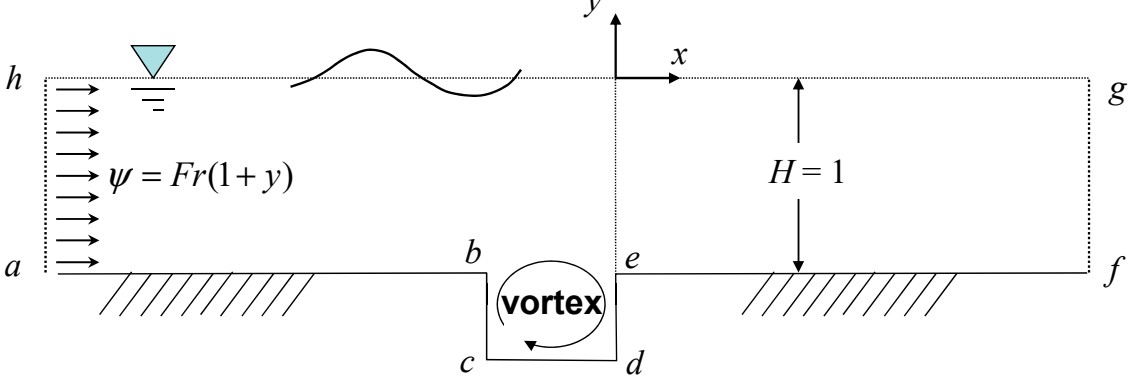

**Figure 1.** Domain configuration and coordinates.

### 2.1. Governing Equations

For an incompressible viscous flow problem in two-dimensional domain, it is convenient to reduce the variables by converting the Navier–Stokes equations into the stream-function–vorticity $(\psi, \omega)$ formulations to describe the flow fields. All variables are written in dimensionless form, using the length scale of undisturbed mean water depth, $H^*$, and respectively the velocity and time scales of $\sqrt{gH^*}$ and $\sqrt{H^*/g}$, with $g$ being gravitational acceleration. The variables with superscript "*" represent the dimensional quantities. A transformation from Cartesian coordinates, $(x, y; t)$, to a boundary-fitted coordinate system, $(\xi, \eta; \tau)$, is carried out by introducing the general transformation $\xi = \xi(x, y)$ and $\eta = \eta(x, y)$. The dimensionless governing equations according to the laws of conservations of mass and momentum can be expressed in terms of vorticity $\omega$ and stream-function $\psi$ in the defined curvilinear coordinate system as

$$\nabla^2 \omega = Re(\omega_\tau + U\omega_\xi + V\omega_\eta) \tag{1}$$

$$\nabla^2 \psi = -\omega \tag{2}$$

The subscripts denote the partial derivatives. Here, $Re = H * \sqrt{gH^*}/\nu$ is a representative Reynolds number, in which $\nu$ is the kinematic viscosity. The symbol $\nabla^2$ is the Laplacian operator, which is defined as

$$\nabla^2 = g^{11}\frac{\partial^2}{\partial \xi^2} + 2g^{12}\frac{\partial}{\partial \xi}\frac{\partial}{\partial \eta} + g^{22}\frac{\partial^2}{\partial^2 \eta} + f^1\frac{\partial}{\partial \xi} + f^2\frac{\partial}{\partial \eta} \tag{3}$$

The variables $U$ and $V$ are the contra-variant components of the relative fluid velocities given as

$$V = (-y_\tau x_\xi + x_\tau y_\xi - \psi_\xi)/J \tag{4a}$$
$$U = (-x_\tau y_\eta + y_\tau x_\eta + \psi_\eta)/J; \quad V = (-y_\tau x_\xi + x_\tau y_\xi - \psi_\xi)/J,$$

$$U = (-x_\tau y_\eta + y_\tau x_\eta + \psi_\eta)/J; \quad V = (-y_\tau x_\xi + x_\tau y_\xi - \psi_\xi)/J, \tag{4b}$$

and

$$g^{11} = (x_\eta^2 + y_\eta^2)/J^2 \tag{5a}$$

$$g^{22} = (x_\xi^2 + y_\xi^2)/J^2 \tag{5b}$$

$$g^{12} = -(x_\xi x_\eta + y_\xi y_\eta)/J^2, \tag{5c}$$

$$f^1 = [(Jg^{11})_\xi + (Jg^{12})_\eta]/J \tag{5d}$$

$$f^2 = [(Jg^{12})_\xi + (Jg^{22})_\eta]/J, \text{ and} \tag{5e}$$

$$J = (x_\xi y_\eta - y_\xi x_\eta) \tag{5f}$$

### 2.2. Initial and Boundary Conditions

A flow field that is approximately confined in a finite domain as indicated by *abcdefgh* in Figure 1 is assumed. The required initial and boundary conditions are described in the following.

### 2.2.1. Initial Condition

At the initial state, an approximated potential flow condition is specified across the entire computational domain. Assuming the free surface is undisturbed and a uniform upstream velocity denoted by $U^*$ is provided to enter at the domain inlet. This quantity is a given parameter and is related to the Froude number as $Fr = U^*/\sqrt{gH^*}$.

### 2.2.2. Boundary Conditions

At the free-surface boundary, defined by $y = \zeta(x, t)$, one of the applicable conditions is the kinematic condition:

$$\psi_\xi + \zeta_\tau x_\xi = x_\tau \zeta_\xi \tag{6}$$

The other is the dynamic free-surface boundary condition:

$$\psi_{\xi\tau}(Jg^{12}) + \psi_{\eta\tau}(Jg^{22}) + \psi_\xi \widetilde{A} + \psi_\eta \widetilde{B} + (u - x_\tau)u_\xi + (v - \zeta_\tau)v_\xi + \zeta_\xi$$
$$+ \omega[(u - x_\tau)\zeta_\xi - (v - \zeta_\tau)x_\xi] + \frac{J}{Re}(g^{12}\omega_\xi + g^{22}\omega_\eta) = 0 \tag{7}$$

in which $(u, v)$ denote the velocity components in the $(x, y)$ directions and are defined as

$$u = (x_\xi \psi_\eta - x_\eta \psi_\xi)/J, \ v = (-y_\eta \psi_\xi + y_\xi \psi_\eta)/J \tag{8}$$

Furthermore, the convective coefficients $\widetilde{A}$ and $\widetilde{B}$ are arranged as

$$\widetilde{A} = -\left(\frac{x_\eta}{J}\right)_\tau x_\xi - \left(\frac{\zeta_\eta}{J}\right)_\tau \zeta_\xi; \ \widetilde{B} = \left(\frac{x_\xi}{J}\right)_\tau x_\xi + \left(\frac{\zeta_\xi}{J}\right)_\tau \zeta_\xi \tag{9}$$

According to Tang [32], the vorticity condition at the free surface $(\omega_f)$ is given as

$$\omega_f = -2\frac{\partial u_f}{\partial \widetilde{n}} \tag{10}$$

where $\widetilde{n}$ is the unit-normal vector towards the interior fluid domain and $u_f$ is the fluid free-surface particle velocity along the tangential direction. In terms of the stream function, it is written as

$$u_f = \frac{g^{22}\psi_\eta + g^{12}\psi_\xi}{\sqrt{g^{22}}} \tag{11}$$

Then, substitution of Equation (11) into Equation (10) yields

$$\omega_f = -2\frac{g^{22}(u_f)_\eta + g^{12}(u_f)_\xi}{\sqrt{g^{22}}} \tag{12}$$

A uniform flow is assumed to proceed from the upstream boundary (domain *ha*), where

$$\psi = Fr(y + 1), \ \omega = 0 \tag{13}$$

The outlet condition at the downstream boundary (domain *fg*) is approximated by the radiation condition, which is expressed as

$$\vartheta_t + (u_{fg} + \sqrt{1 + \zeta})\vartheta_x = 0 \tag{14}$$

where $\vartheta$ is a dummy variable, representing either $\psi$, $\omega$, or $\zeta$. The $u_{fg}$ denotes the downstream horizontal velocity, which is numerically evaluated using the Neumann condition. The impermeable solid boundary (domain *abcdef*) resembles as a streamline, which can be specified with a constant stream function value, i.e., $\psi = 0$. The no-slip condition along cavity bottom is applied to the derivation of the impermeable surface vorticity. A standard formulation is taken from Nallasamy [33]. On the bottom, the wall vorticity is expressed as

$$\omega = -2\psi_1/\Delta n^2 \tag{15}$$

Here, $\psi_1$ represents the stream function value at the first grid node adjacent to the wall in the fluid domain and $\Delta n$ is the normal distance between the wall and the adjacent node.

## 3. Numerical Method

### 3.1. Grid Generation and Discretization

In the numerical method, the algebraic grid-generation technique is adopted for curvilinear grids generated to fit the free surface, as it undergoes deformation. The indices of grid points are denoted by $i = 1$ to *IM* within the computational domain $(x_{min}, x_{max})$ and $j = 1$ to *JM* within $(y_{min}, y_{max})$ in the $x$- and $y$-directions, respectively. The size of the mesh in the $x$ direction is fixed, however, distributed nonuniformly in space. As shown in Figure 2, the domain along the $x$ direction is divided into three regions: the first region extends from $i = 1$ to $i = i_c$, the second region covers the cavity from $i = i_c$ to $i = i_{c1}$, and the indices from $i = i_{c1}$ to $i = IM$ are arranged for the third region. However, along the $y$ direction, the domain is divided by the line numbering $j = j_c$ into two parts: the lower region measures from $j = 1$ to $j = j_c$ and the upper one extends from $j = j_c$ to $j = JM$. The grid size in $y$ direction below $y_{jc}$ is fixed while in the upper region (above $y = y_{jc}$) it is transiently evolved with the varying free surface. In order to generate the numerical grid efficiently, the following non-uniform grid system is adopted:

$$x_i = x_{i_c} - \frac{a_0(1 - r_L^{i_c - i})}{1 - r_L}, \text{ for } i = 1 \text{ to } i_c \tag{16a}$$

$$x_i = x_{i_c} + \frac{(x_{i_{c1}} - x_{i_c})}{i_{c1} - i_c}(i - i_c), \text{ for } i = i_c \text{ to } i_{c1} \tag{16b}$$

$$x_i = x_{i_{c1}} + \frac{a_0(1 - r_R^{i - i_{c1}})}{1 - r_R}, \text{ for } i = i_{c1} \text{ to } IM \tag{16c}$$

$$y_j = y_1 + \frac{(y_{jc} - y_1)}{j_c - 1}(j - 1), \text{ for } j = 1 \text{ to } j_c \tag{16d}$$

$$y_{i,j} = y_{jc} + \frac{\zeta_i - y_{jc}}{JM - j_c}(j - j_c), \text{ for } j = j_c \text{ to } JM \tag{16e}$$

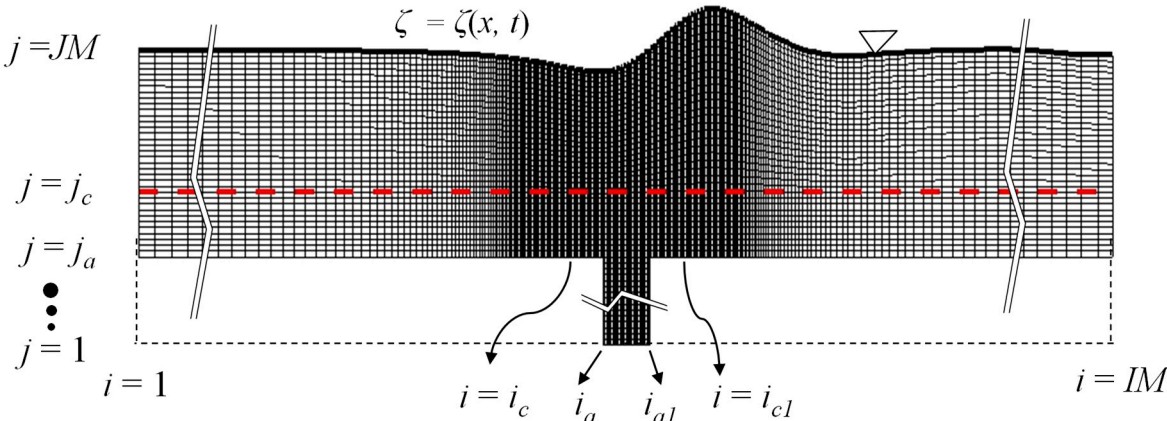

**Figure 2.** Schematic of grid structure.

It should be noted the grid points including $j = 1$ to $j = j_a$ indices are assigned only inside the cavity with $x$ indices covering from $i = i_a$ to $i = i_{a1}$. An example plot showing a grid system covering the fluid domain, including the free surface and a cavity, is illustrated partially in Figure 2. For this case, the parameter values are set as $a_0 = 0.02$, $r_L = 1.0451$, $r_R = 1.0426$, $i_c = 101$, $i_a = 151$, $i_{a1} = 201$, $i_{c1} = 251$, $j_a = 51$, $j_c = 76$, $x_{i_c} = -2$, $x_{i_a} = -1$, $x_{i_{a1}} = 0$, $x_{i_{c1}} = 1$, $y_1 = -2$, and $y_{j_c} = -0.5$. This gives a grid system with equal spacing along x and y directions within the cavity, where $\Delta = \Delta x_{cavity} = \Delta y_{cavity} = 0.02$. The grid sizes of $\Delta = 0.05$ and $\Delta = 0.01$ have also been selected to test the effect of grid size on the results of the flow condition $Fr = 1.0$ and $Re = 5000$. To analyze the grid-size influence on the evolving vortices, results obtained from using the three different mesh sizes (not shown here) are compared. It is found the vortical flow patterns in a larger scale are generally similar to those from the setup of any of the three different meshes, however, with the differences shown on the corner eddies—where they cannot be revealed in the domain using coarse grid ($\Delta = 0.05$). On the other hand, the simulation in a domain with a finer grid size of $\Delta = 0.01$ costs vast computational efforts to obtain the similar results as those from the case of $\Delta = 0.02$. Therefore, the grid size of $\Delta = 0.02$ within the cavity is utilized to solve all cases in the present study.

The finite analytic method (FAM) developed by Chen and Chen [34] is applied to solve the coupled system of Equations (1) and (2) in the flow domain. Recent applications of the FAM based SVFS model have been addressed in Chang et al. [31] and Chang and Lin [35]. The focus described here is to provide a new numerical scheme applied to solve the boundary conditions. Numerically, it is more challenging in solving the free-surface conditions. An upwind scheme for the free-surface computation is employed. Details of the numerical treatment of this model are described in Appendix A. Once the free-surface variables $\psi$ and $\zeta$ are determined, the grid points are regenerated, and the time-marching solutions with increment $\Delta\tau = 0.01$ are obtained by satisfying the governing equation and other associated boundary conditions. The converging criteria for both $\psi$ and $\zeta$ are reached when the absolute deviations of the iterated variables are less than $10^{-6}$, and $\omega$ is less than $10^{-4}$.

### 3.2. Numerical Procedure

The numerical procedure of solving the above-described discretizations for obtaining the converged solutions at each time step is given below.

Initially (at $t = 0$), a still water surface is assumed. The initial values of $\zeta$ and $\omega$ are zero. Then, the computational grids are generated according to the grid formulations (Equation (16)) and the grid metric coefficients in Equation (5) are determined. Initial values of $\psi$ in the whole flow domain are calculated by solving Equation (2) with the settings of $\omega = 0$, the inlet uniform flow, and other boundary conditions.

With the values obtained for $\psi_{i,j}^n$, $\omega_{i,j}^n$, and $y_{i,j}^n$ (including $\zeta_i^n$) at the time level $n$, the computation of the unknown variables at the new time level ($t = (n+1)\Delta t$, $n = 0, 1, 2, 3 \ldots$ ) follows the steps as shown below:

1. Solve the kinematic free-surface boundary condition (Equation (A2)) to obtain $\hat{\zeta}_i^{n+1}$, and accordingly update $\hat{y}_{i,j}^{n+1}$ from Equation (16e). (Here, the notation of hat "^" represents the provisional solutions).

2. Solve the dynamic free-surface boundary conditions (Equations (A3) and (12)) for the values of $\hat{\psi}_{i,JM}^{n+1}$, and $\hat{\omega}_f^{n+1}$, respectively.

3. Update the wall vorticity from Equation (15).

4. Regenerate the vertical coordinates $\hat{y}_{i,j}^{n+1}$ (Equation (16)) and calculate the grid metric coefficients (Equation (5)).

5. Solve the coupled Equations (1) and (2) to obtain $\hat{\psi}_{i,j}^{n+1}$ and $\hat{\omega}_{i,j}^{n+1}$, respectively, in the flow field.

6. Repeat steps 1–5 until converged solutions are obtained.

The numerical procedure of the above six steps is then carried into the next time step, and the calculation is continued until the final allotted time is reached.

## 4. Results

### 4.1. Model Validations

Model validations for the computed free-surface elevations are conducted by comparing the results with published solutions for the cases of a potential uniform flow passing over a bottom-depressed region. In addition, the vortex evolution is validated by the comparisons with the results of a lid-driven cavity flow without a free surface.

#### 4.1.1. Free-Surface Wave Generation Due to a Negative Bottom Forcing Function

The free-surface deformations generated due to a flow passing over a bottom hump-like structure have been frequently studied. For example, Lowery and Liapis [36] studied the wave motions produced by a flow passing over a bottom-mounted, semi-circular cylinder. The hump case is referred to as having a positive forcing effect, and in the presence of a depression, it represents a negative forcing function (Wu, [35]). The free-surface motions resulting from the input of a negative forcing function are very different from those under the condition of a positive forcing function. In general, it is also more difficult to achieve stable solutions in computations. Zhang and Chwang [20] investigated the different wave systems due to the consideration of either a positive or a negative forcing function. They solved for the primitive variables in the Euler equations using the finite difference method. Here, one of their cases is selected for results comparisons. As shown in Figure 3, the bottom shape is expressed by

$$b(x) = \frac{b_m}{2}\left[1 + \cos(\frac{2\pi}{L})\right], \; -L/2 \le x \le L/2, \tag{17}$$

The selected flow condition is the same as in [20] with the inputted parameters of $Fr = 1.0$, $L = 2.0$, and $b_m = -0.1$. In our numerical calculation, the primary variables (stream function and free-surface elevation) in this potential flow problem are analyzed for their convergence. As shown in Figure 4. The time-step convergence requirement is that the absolute difference between the previous value and the current value must be $< 10^{-6}$. It can be seen from the plot that the convergence of the stream function is slower than that of the free-surface elevation, although convergence is reached eventually.

The results showing the successive free-surface profiles from $t = 0$ to 400 are compared in Figure 5a,b. Comparing the results shown in Figure 5a,b, it is noted the numbers and the temporal variations of the advancing upstream solitons obtained from the present model are in good agreement with those from Zhang and Chwang [20], although with a similar pattern the present model produces slightly stronger trailing dispersive waves downstream of the structure. It is demonstrated through this comparison study that the second-order upwind scheme is suitable to be applied for all cases that follow.

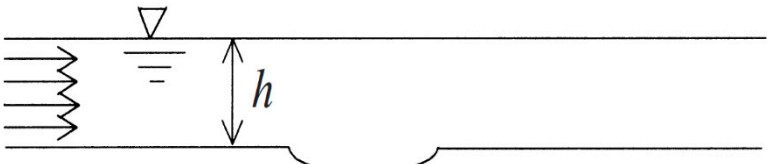

**Figure 3.** Diagram of the flow over a depression.

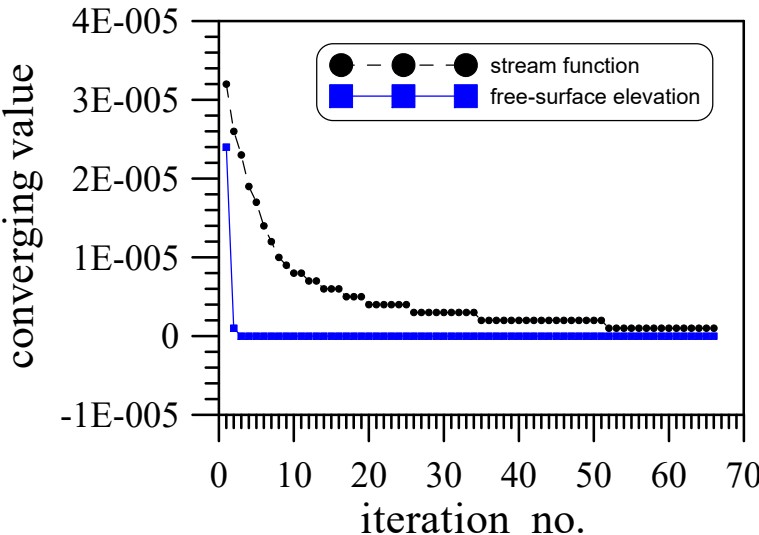

**Figure 4.** Processes of a typical time step of convergence iterations for the primary variables.

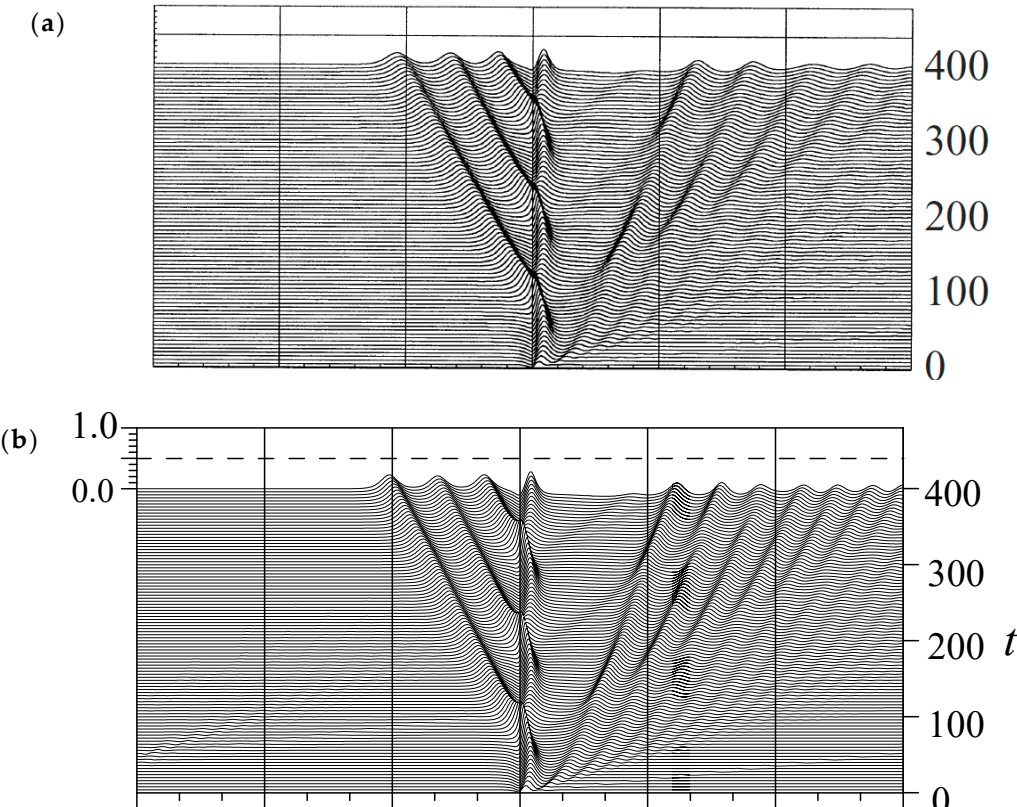

**Figure 5.** Free-surface evolution over a negative bottom: (**a**) Zhang and Chwang, adapted from [20], with permission from Copyright © 2001 Cambridge University Press; (**b**) present results.

### 4.1.2. Vortex Motion Comparison: Pure Lid-Driven Cavity Flow

The laboratory data or other numerical results for vortex motions induced by uniform flows with a free surface over a cavity are not currently available. Therefore, a lid-driven flow in a closed cavity, as shown in Figure 6, is tested for comparisons with other studies. The flow at the lid is assumed to move with a constant horizontal velocity while the flows at other walls are stationary. The Reynolds number for the cavity flow is defined as $Re_{lid} = u^*_{lid} L^* / v$, in which $u^*_{lid}$ is the velocity at the lid and $L^*$ is the length of the cavity. Hence, the lid-normalized velocity $u = u_{lid} = 1$. Traditionally, the quantitative

analyses are made to compare the velocity profiles in the cavity. In case of steady flow, the x-component of velocity (u) and the y-component of velocity (v) are both plotted along the y-axis at x = 0.5 and the x-axis at y = 0.5, respectively. It is compared for five set grids (101 × 101, 121 × 121, 151 × 151, 181 × 181, and 201 × 201) having different numbers of grid nodes for calculating the flow field, as denoted by the velocity profiles presented in Figure 7. In this figure, it can be qualitatively seen that 181×181 has almost reached grid independence. It is conservatively chosen the results of 181 × 181 to compare them with the results of Ghia et al. [37], where a clearer plot of their results was given in Peng et al. [38].

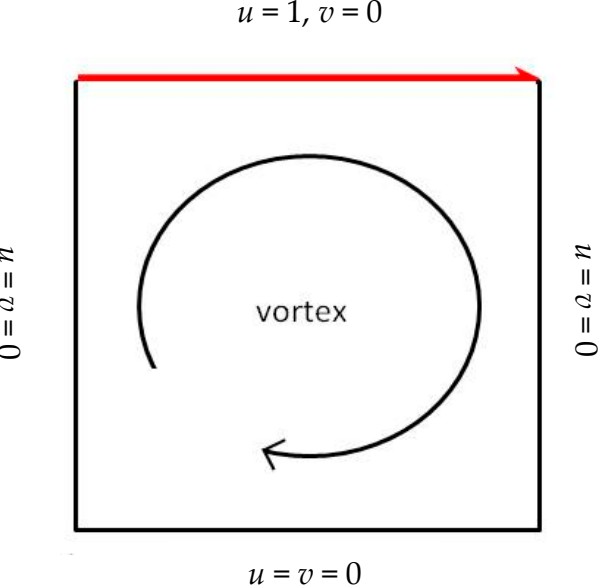

**Figure 6.** Sketch of a cavity domain with a constant speed at the lid.

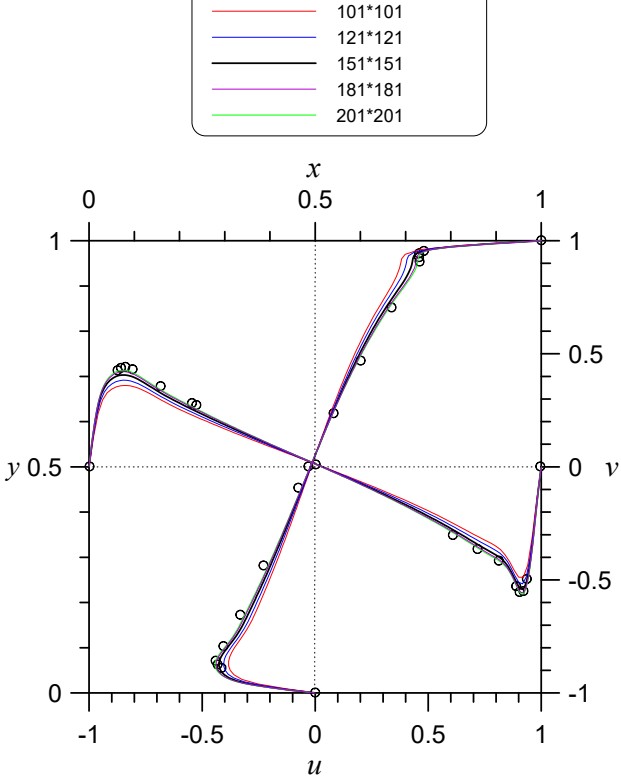

**Figure 7.** Computed steady velocity profiles for $Re_{lid}$ = 5000.

Figure 8 depicts the cavity flow phenomenon with streamline patterns for a selected flow condition, $Re_{lid}$ = 5000. The qualitative comparisons with other studies reveal the achievement of similar patterns. In addition to show the present model results (Figure 8a), plots of the numerical solutions of Ghia et al. [37] and Erturk et al. [39] are presented in Figure 8 for comparisons. The present solutions employ a uniform grid system of 181 × 181, while Ghia et al. [37] used 257 × 257 nodes, and the simulation of Erturk et al. [39] used a 129 × 129 grid system. The solutions from the present investigation are drawn with streamline contours at the interval of 0.01 for the values ranging from −0.11 to 0 and 0.0002 for the values between 0 and 0.048. In this case, the flow cannot reach a steady state until after $t$ = 40. The results at $t$ = 300 are shown for comparisons to ensure that the steady-state situation has been reached.

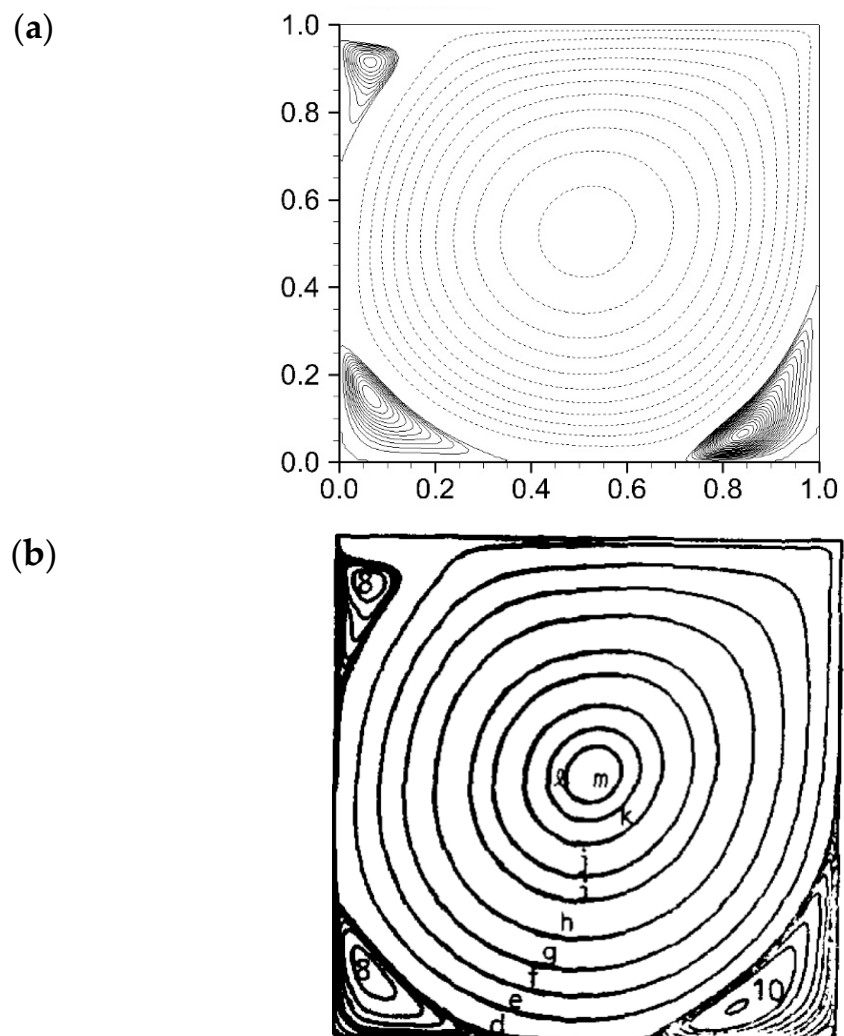

**Figure 8.** *Cont.*

(c)

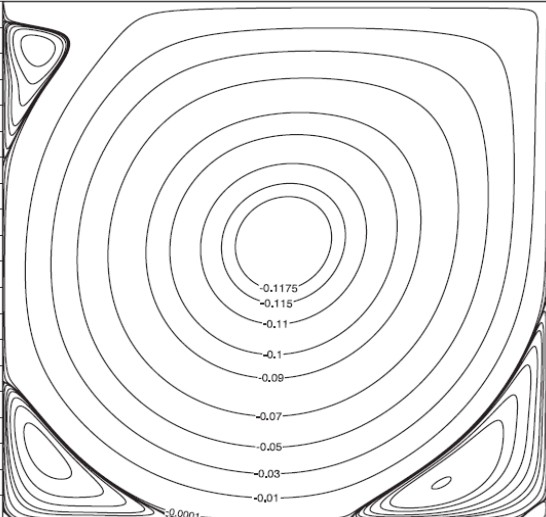

**Figure 8.** Comparisons of streamline patterns for Re$_{lid}$ = 5000 in a square cavity. (**a**) Present solutions; (**b**) Ghia et al., adapted from [37], with permission from Copyright © 1982 Published by Elsevier Inc.; (**c**) Erturk et al., adapted from [39], with permission from © Mar 11, 2005, John Wiley & Sons Ltd.

### 4.2. Vortical Flows in a Square Cavity with a Free Surface: Fr = 1.0, Re = 5000 and 500

In this subsection, we discuss the vortical-flow structures at various values of *Re*. For an inlet flow with a constant speed, *Fr* = 1.0, the phenomena of generated vortices for flows passing over a 1 × 1 square cavity at two Reynolds numbers, *Re* = 5000 and 500, are considered.

The case with *Re* = 5000 is first discussed for the streamline patterns in Figure 9. During the transient process of vortex evolution, it reveals that a shear flow separating the left corner forms a clockwise vortex (Figure 9a) immediately and enlarges to occupy almost the entire cavity space at *t* = 2 (Figure 9c). As time progresses, a small counterclockwise eddy emerges from the right wall (Figure 9d). This secondary vortex continues to expand to fill nearly half of the space in the lower right part of the cavity (Figure 9e). Later, this bottom counterclockwise vortex transitions to the left wall and extends upwards to reduce the region of the clockwise vortex on top of it (Figure 9j). With the further developments of the vortices under the influence of a near-constant shear-flow velocity passing over the opening of a cavity, as shown in Figure 9p, the streamline patterns show a flattened, oval, clockwise vortex that is visible at the top part of the cavity and appear below it is a large counterclockwise, but weak vortex. It should be noted this counterclockwise vortex is evolved from a secondary vortex. In addition, small eddies, which are too weak to be displayed clearly, appear at both bottom corners. This pattern is different from that of a pure cavity flow at *Re$_{lid}$* = 5000, which can eventually reach a steady-state flow condition. In Figure 9, the free surface evolves with time, yielding a movable cavity lid that interacts with the fluid inside and outside of the cavity.

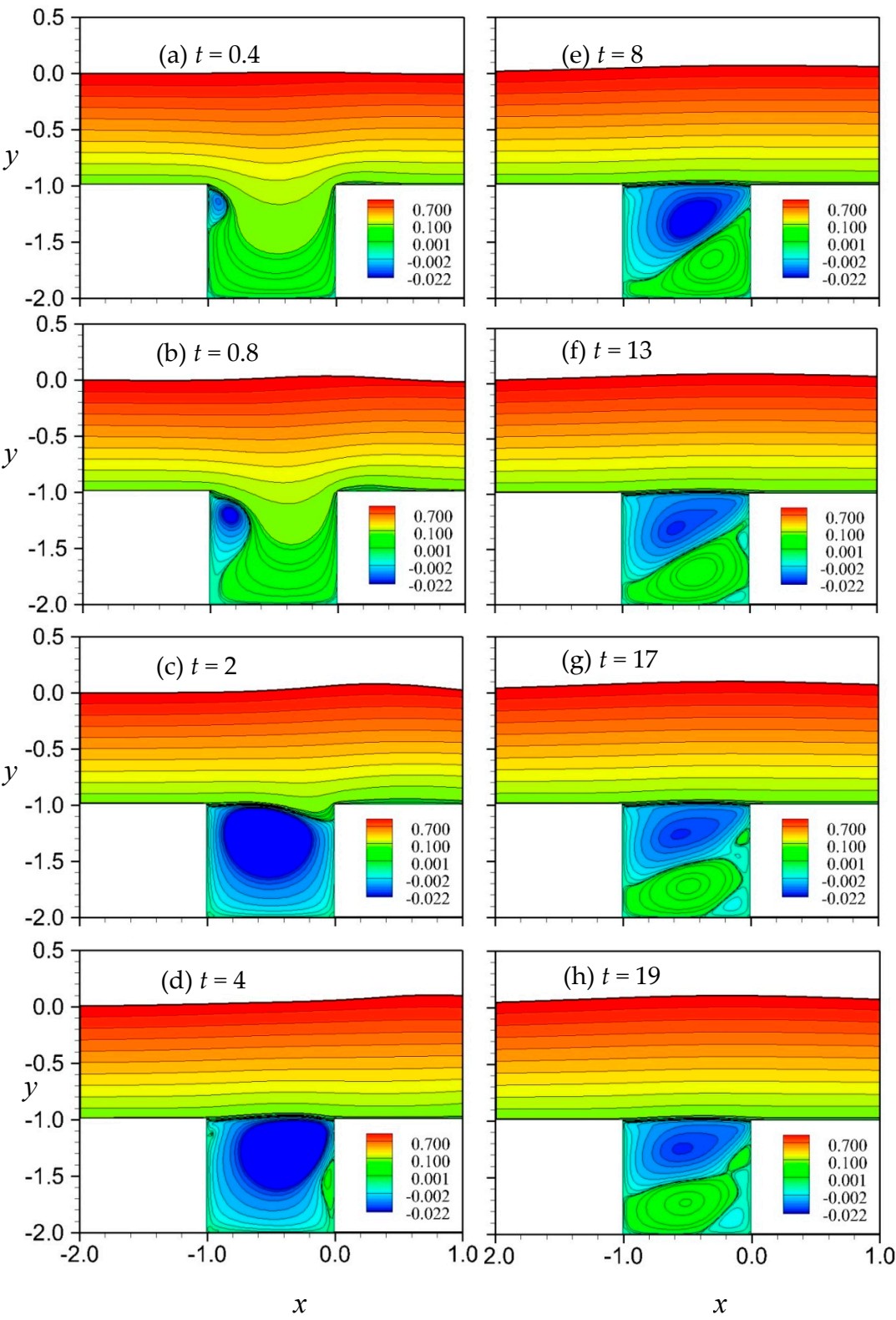

**Figure 9.** *Cont.*

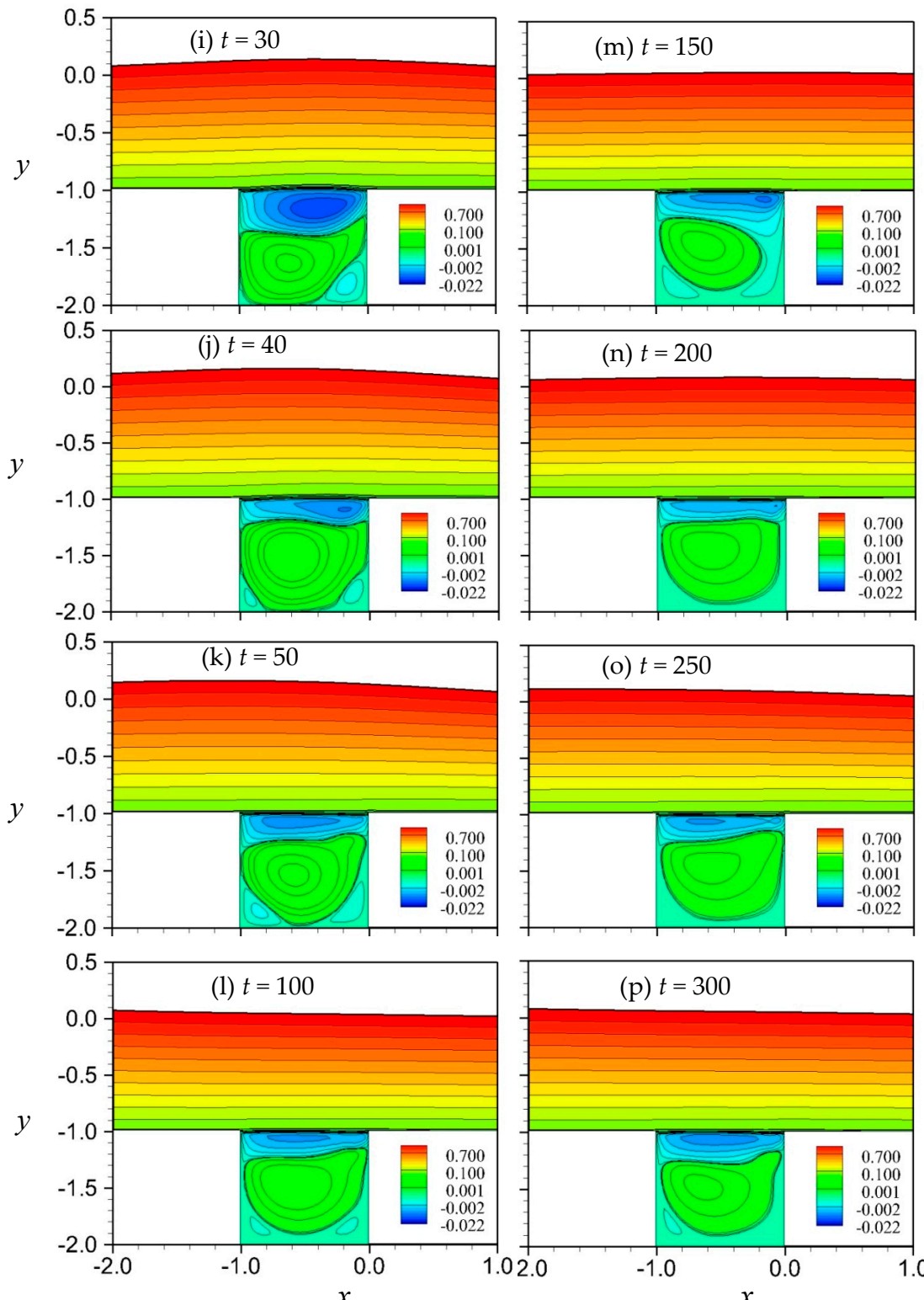

**Figure 9.** Streamline patterns in a square cavity for $Re$ = 5000 and $Fr$ = 1.0 at various phases: (**a**) $t$ = 0.4, (**b**) $t$ = 0.8, (**c**) $t$ = 2, (**d**) $t$ = 4, (**e**) $t$ = 8, (**f**) $t$ = 13, (**g**) $t$ = 17, (**h**) $t$ = 19, (**i**) $t$ = 30, (**j**) $t$ = 40, (**k**) $t$ = 50, (**l**) $t$ = 100, (**m**) $t$ = 150, (**n**) $t$ = 200, (**o**) $t$ = 250, and (**p**) $t$ = 300.

For the case of the lower Reynolds number, $Re$ = 500, Figure 10 illustrates the time variations of the vortex flows in a cavity and the near steady situation is achieved after about $t$ = 30 (Figure 10g). The main vortex pattern is clearly shown to be different from that presented in Figure 9. In this case,

an obvious eddy occurs at the bottom right corner (Figure 10d). A small but visible eddy appears at the left corner. This feature is similar to the pattern of pure cavity flow for a lower value of $Re_{lid}$ described in Ghia et al. [37]. It is worth to note, although the flow pattern inside the cavity seems to reach a near steady state, the surface waves are still in a transient condition.

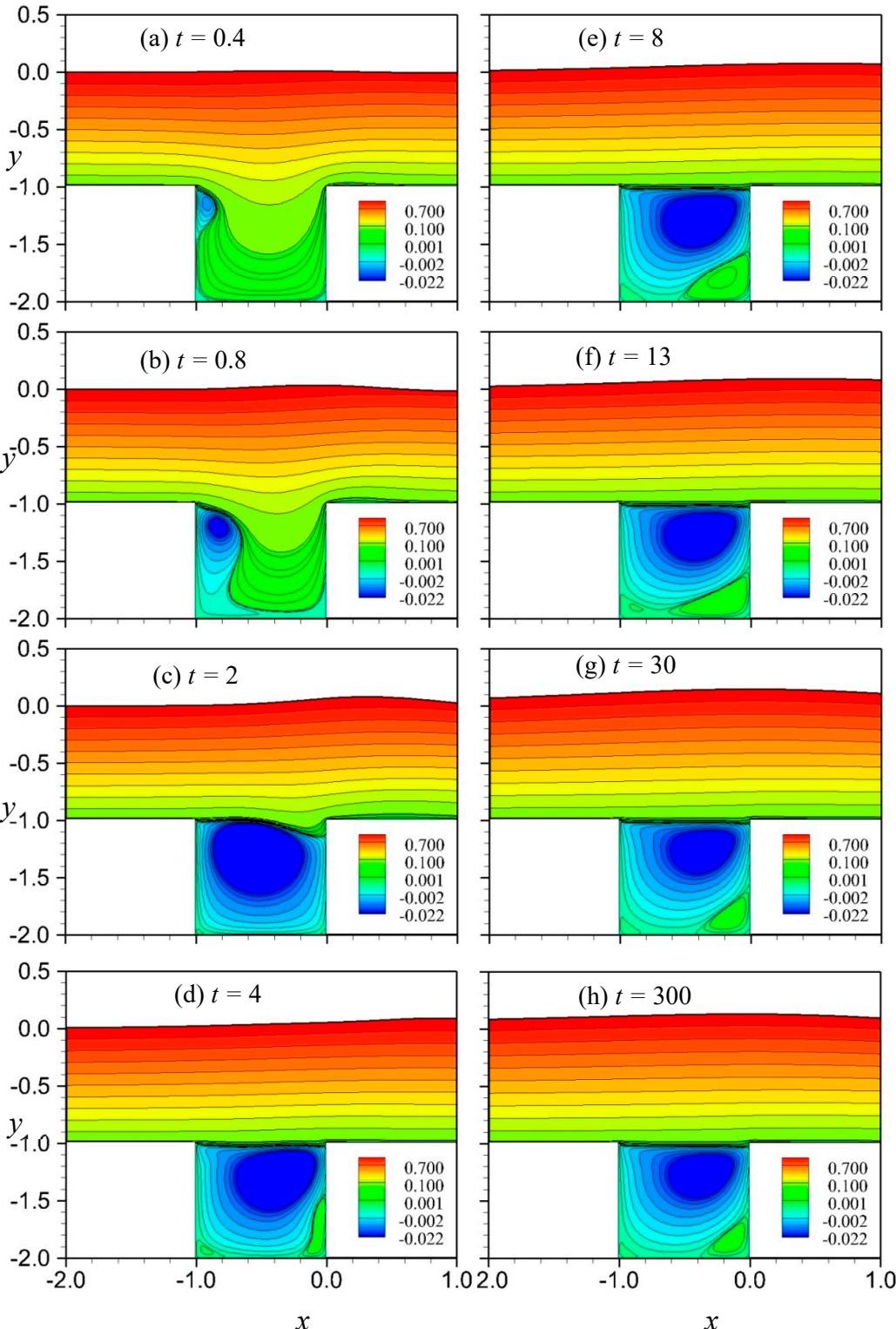

**Figure 10.** Streamline patterns in a square cavity for $Re = 500$ and $Fr = 1.0$ at various phases: (**a**) $t = 0.4$, (**b**) $t = 0.8$, (**c**) $t = 2$, (**d**) $t = 4$, (**e**) $t = 8$, (**f**) $t = 13$, (**g**) $t = 30$, and (**h**) $t = 300$.

To examine the effect of the Reynolds number on the vortical flow, the viscous flow patterns at $t = 12$ for the cases of $Re = 5000$ and 500 are compared with each other as well as with the results from the potential flow in Figure 11, where the inflow condition is set as $Fr = 1.0$. As can be seen in Figure 11a, the potential flow solutions show no vortices produced in the cavity; which is not realistically reflected in nature. The fluid accelerates in the upward direction downstream of the cavity to force the free surface to rise up violently and eventually reach a condition violating the potential flow theory. For the viscous fluid flow cases, the cavity is almost completely occupied by vortices. For $Re = 5000$, Figure 11b reveals that the flow structure in the cavity is in the developing stage and the space is occupied mainly by a secondary vortex (green part). It finally reaches a quasi-steady like flow (see Figure 9). For $Re = 500$, the primary vortex (blue part, see Figure 11c) is always dominant in the cavity and a steady-state flow can be formed eventually.

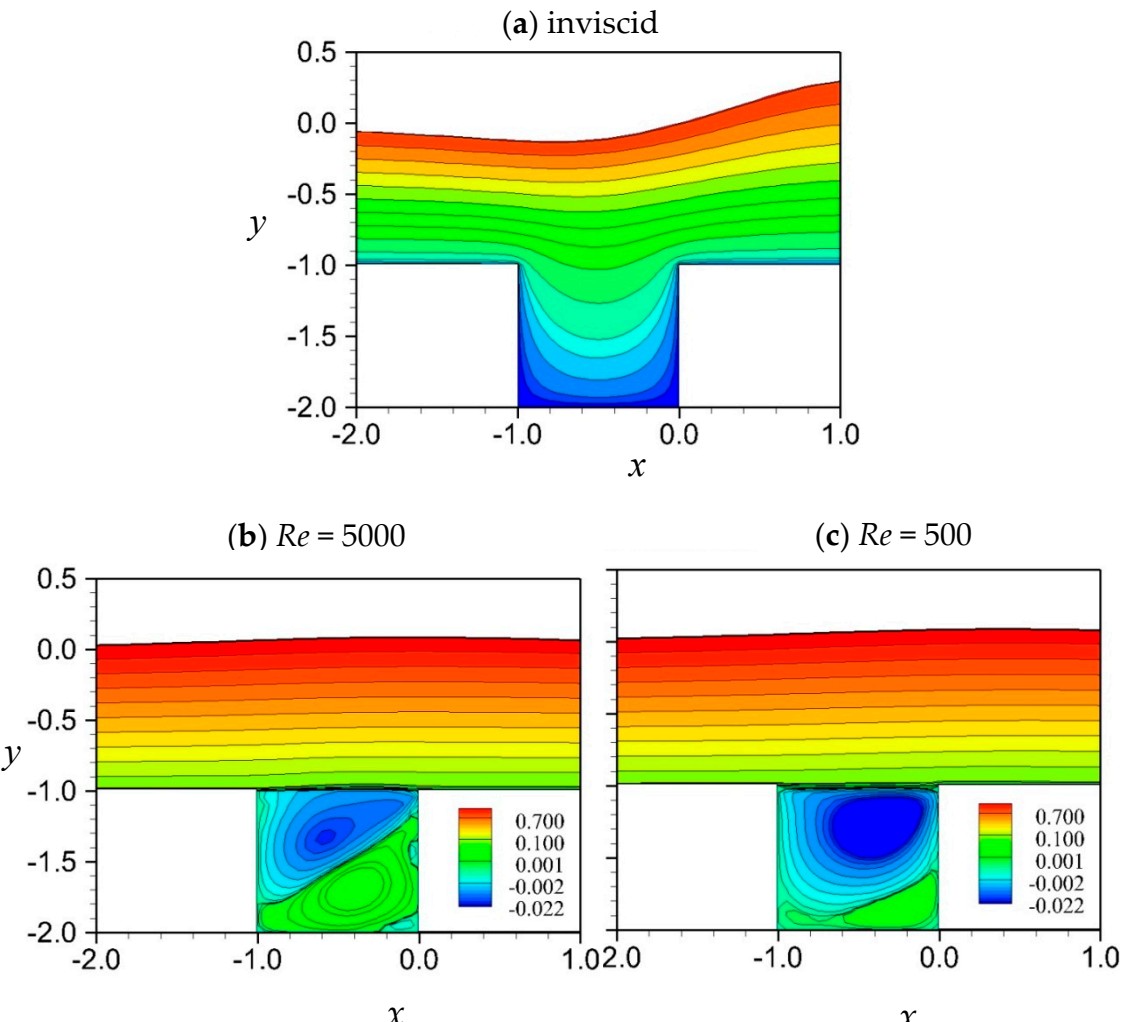

**Figure 11.** Streamlines for $Fr = 1.0$ at $t = 12$: (**a**) inviscid flow, (**b**) $Re = 5000$, and (**c**) $Re = 500$.

*4.3. Free-Surface Elevations for a Square Cavity at Re = 5000 and 500 with Fr = 0.5 to 1.1*

To analyze the influence of the Froude number on the free-surface motion, the Reynolds number is fixed at $Re = 5000$ or 500. The water surface elevations at the central cavity position are plotted in time histories ($t = 0$ to 300) for $Fr$ ranging from 0.5 to 1.1 in Figure 12. The flow ranges cover the subcritical, critical, and supercritical regions. The varying displacement of water surface reflects the potential formation of advancing waves for a specified $Fr$. The tendencies of free-surface pattern are similar between the plots for the cases of $Re = 5000$ and 500 but vary in phase and strength. For a

subcritical flow condition (e.g., *Fr* = 0.5), it appears the free surface is reflected with the small undular motions. With an increase of *Fr*, the forms of waves are enhanced and become stronger (see Figure 12). However, as it has not ever been reported for the case of *Fr* ≈ 1.2, this higher supercritical flow may cause unstable situation. This transition regime at 1.1 < *Fr* < 1.2 was ascertained by Lee et al. [40] in analyzing a hump forcing function. They also indicated that for the positive forcing cases the wave breaking was experimentally observed to occur in the advancing solitary waves at lower supercritical speeds (about 1.1 < *Fr* < 1.2).

(**a**) *Re* = 5000

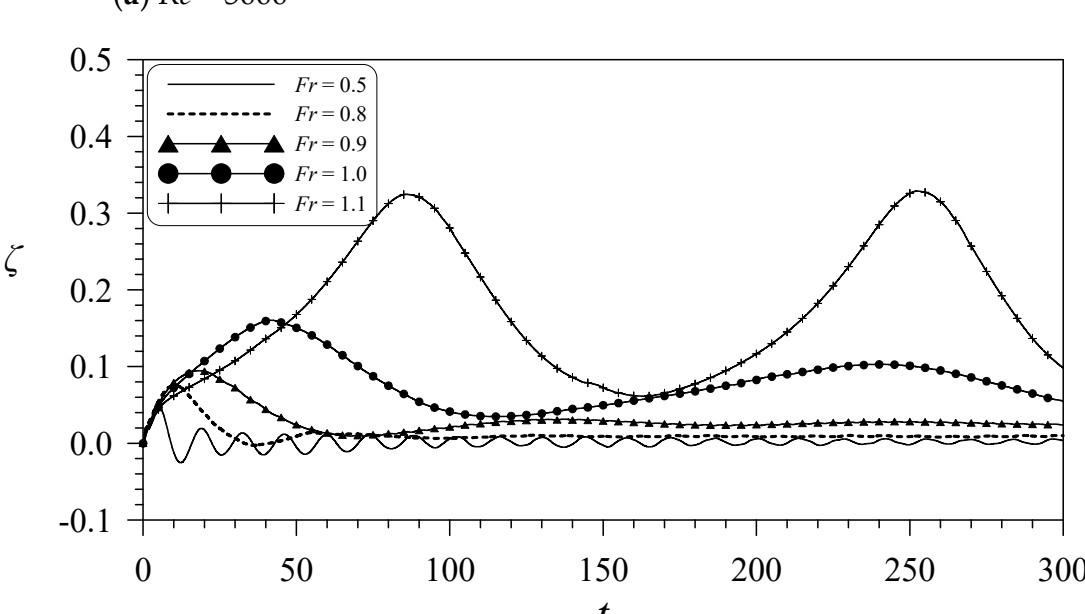

(**b**) *Re* = 500

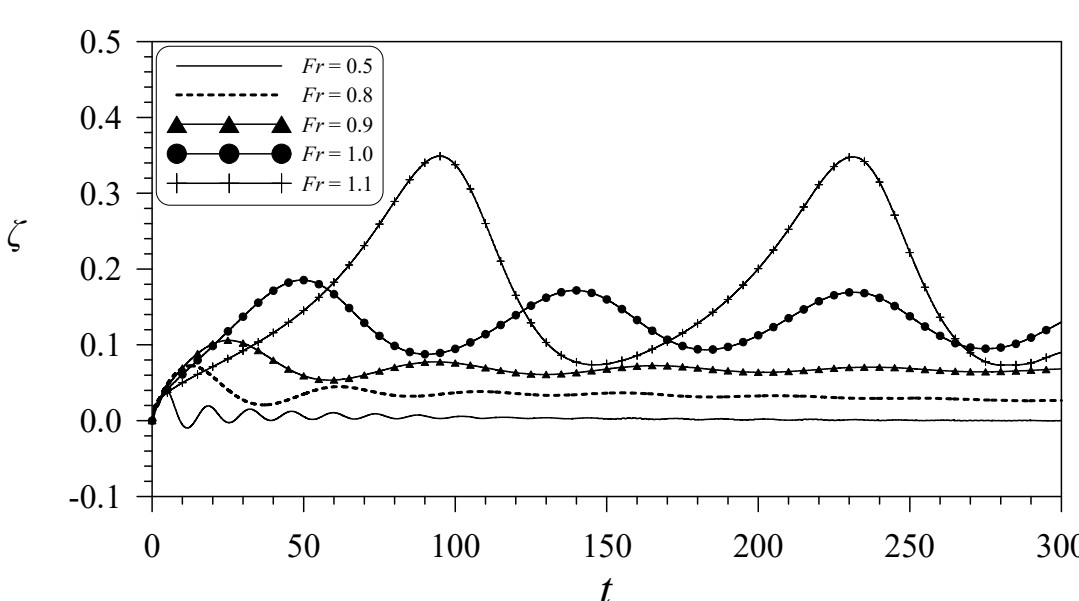

**Figure 12.** Free-surface elevation at *x* = −0.5 for various *Fr* at (**a**) *Re* = 5000 and (**b**) *Re* = 500.

Figure 13 shows the subplots of the time varying free-surface elevations for various values of *Fr* to reflect the differences of the water surface between the conditions of *Re* = 5000 and 500. For the

range of near-critical flow, $Fr$ = 0.8, 0.9, 1.0, and 1.1, the waves are large enough to show the obvious viscous influences; including the period between each of the upstream advancing solitons. For the lower value of $Re$, the wave elevation is higher and the period is shorter. By contrast, the viscous effect on the lower subcritical flow (e.g., $Fr$ = 0.5) is inconspicuous.

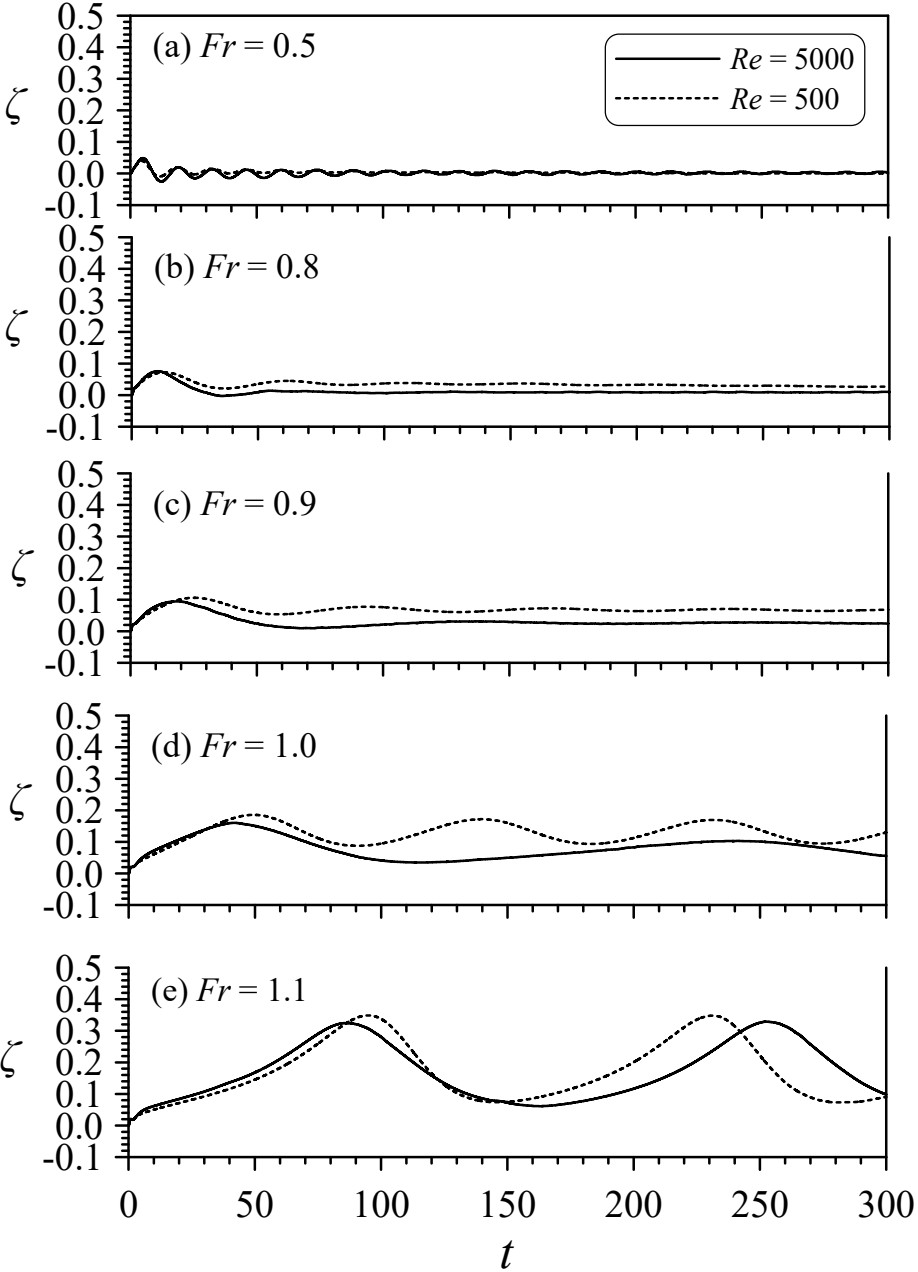

**Figure 13.** Time varying free-surface elevations at $x$ = −0.5 for $Re$ = 5000, 500, and at various $Fr$: (**a**) $Fr$ = 0.5, (**b**) $Fr$ = 0.8, (**c**) $Fr$ = 0.9, (**d**) $Fr$ = 1.0, and (**e**) $Fr$ = 1.1.

*4.4. Free-Surface Profiles at Various Fr for Re = 5000 and 500*

With the same conditions as in Figure 13, the results of the spatial wave profiles at $t$ = 300 are plotted in Figure 14. The basic phenomena of the wave pattern produced by transcritical flow over a disturbance have been described explicitly by Lee et al. [40]. Similar undular bores are found in Figure 14d with $Fr$ = 1.0 where the indicated three flow regions can be noticed. It is observed that there are three separate solitons generated when $Re$ = 500. For the case of higher-subcritical flow (e.g.,

*Fr* = 0.9), the trailing waves are stronger while the amplitude of advancing solitons become larger for lower-supercritical flow (e.g., *Fr* = 1.1). Comparing the advancing solitons generated under the conditions of *Re* = 5000 and *Re* = 500, the wave displacement for *Re* = 5000 is found to be less than that for *Re* = 500. This is potentially a result of the thicker boundary layer from the condition of lower *Re* allowing the additional strength and the disturbance to be extended to a greater region to produce stronger wave motions. In spite of this, after a long traveling distance, the waves are expected to undergo more damping for a lower value of *Re*. It is also interesting to note the wave motions are affected by the movement of the lid streamline of the cavity as it interacts with the inlet flow to influence the wave patterns. It was found that when *Re* = 500, the separation streamline lid in the cavity could reach steady state. However, the separation-streamline lid of *Re* = 5000 showed a slightly movable boundary. Overall, the lid of *Re* = 500 was lower than the average of *Re* = 5000. The lower lid of Re = 500 will be shown to be one of the causes of large wave fluctuations.

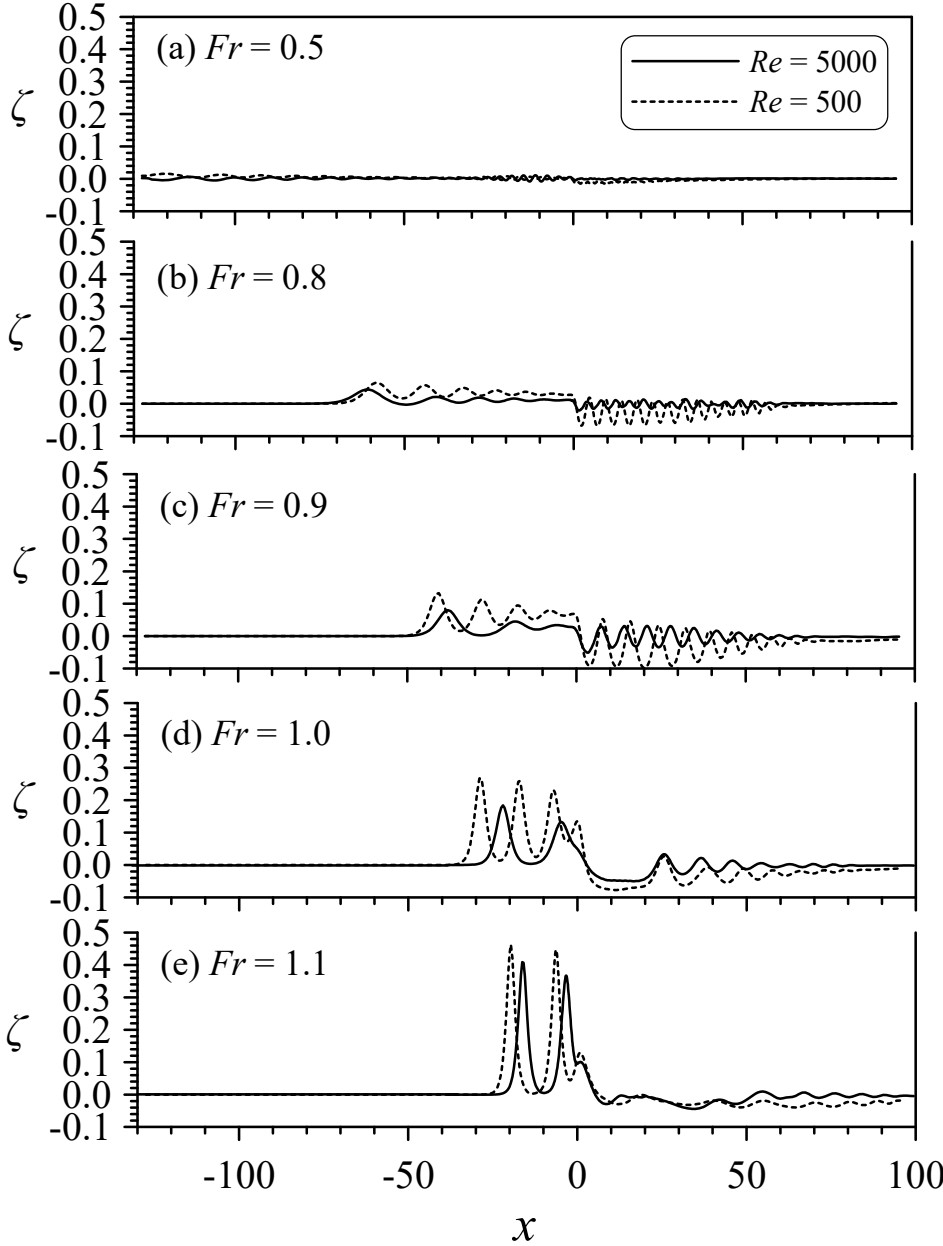

**Figure 14.** Comparisons of the free-surface profiles at *t* = 300 for *Re* = 5000 and 500 with various *Fr*: (**a**) *Fr* = 0.5, (**b**) *Fr* = 0.8, (**c**) *Fr* = 0.9, (**d**) *Fr* = 1.0, and (**e**) *Fr* = 1.1.

The near-critical flows create dramatically different wave phenomenon, especially the upstream advancing solitons. Similar phenomena were found in the near-critical flow over a hump in Lee et al. [40]. They addressed both the amplitude of the upstream-advancing waves and their generation period increase very rapidly as *Fr* approaches the limiting value of about 1.2. This was also one of the incentives behind this article. The flow pasts over a protrusions can produce this phenomenon. So what about the cave and the influence of fluid viscosity? To illustrate the process of the wave formation, the free-surface profiles are distinctly plotted with perspective view plots in the *x-t* plane, as shown in Figure 15. Only the case of *Re* = 500 is selected to plot the wave elevations under the transcritical flow conditions, *Fr* = 0.9, 1.0, and 1.1. In each subplot, the free-surface profiles are displayed from *t* = 0 to 300 at an interval of three unit times. Similar to the explanations mentioned above, but with more insights, the results show the dramatic differences of upstream and downstream wave patterns among the cases from high subcritical *Fr* (e.g., *Fr* = 0.9) to low supercritical (e.g., *Fr* = 1.1). The wave trough/peak traces in *x-t* plane implies the swift movement of the generated waves. The wave patterns of near-critical flows passing over a bottom cavity are similar to those of flows over a hump. The reason is the depressed region will be filled with vortices and a movable cavity lid is formed above the cavity. The movable lid plays a similar role as the hump to produce the surface disturbance for the advancement of a group of solitary waves upstream of the cavity. In terms of the free-surface elevations behind the cavity, the high subcritical flow (e.g., *Fr* = 0.9), when compared to the low supercritical flow, tends to make the downstream oscillatory trailing waves with higher frequency and amplitude resembling a train of modulated cnoidal waves. Also, its upstream undular bores are weaker. In contrast, the low supercritical flow (e.g., *Fr* = 1.1) produces a group of higher and clearly separated solitons but weaker trailing waves with longer periods resembling indeed small-amplitude long waves.

### 4.5. Influence of Cavity Depth

This section will discuss the effects of the depth of the cavity. Consider Re = 5000, as an example. Figure 16 denotes the stream patterns in the cavity with different aspect ratios (shallow half) when t = 300. Their vortex motion can reach the quasi-steady state. In addition, both of them also exhibit a flat elliptical eddy in the cavity near the lid. Figure 16a is a shallower-depth cavity. The virtual cavity lid of the shallower one is slightly above −1. Also, to compare their wave elevation at a fix position (*x* = −0.5), Figure 17 shows the shallower cavity makes the shorter wave period and larger waves. The vortex motion in cavity will play a very important role to influence the wave evolutions. The cavities of different depths or shapes will affect the vortices and the wave motions. The more details for the effects of geometry and size of cavity may be discussed in our future work.

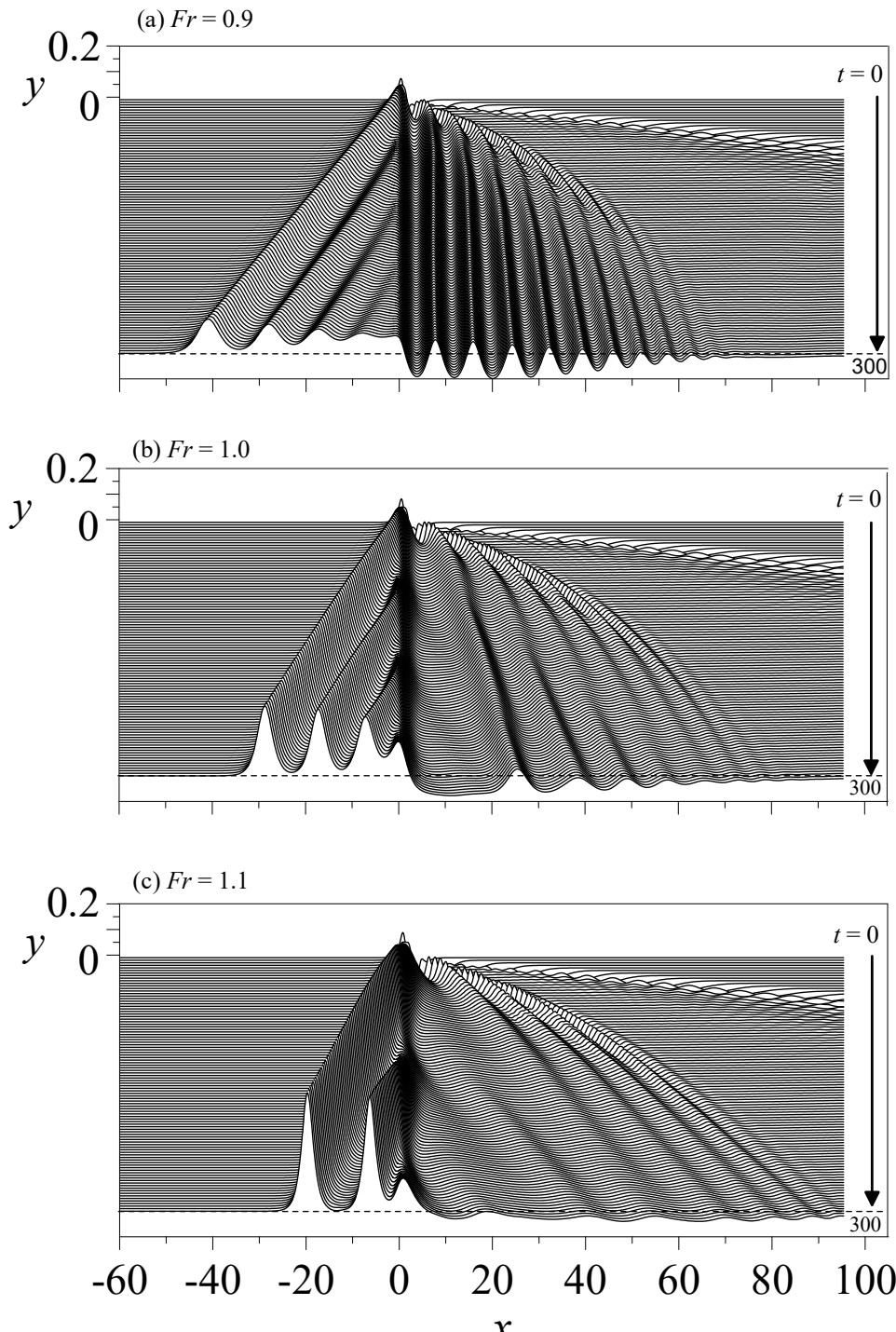

**Figure 15.** Time series plots of wave elevations at *Re* = 500 for (**a**) *Fr* = 0.9, (**b**) *Fr* = 1.0, and (**c**) *Fr* = 1.1.

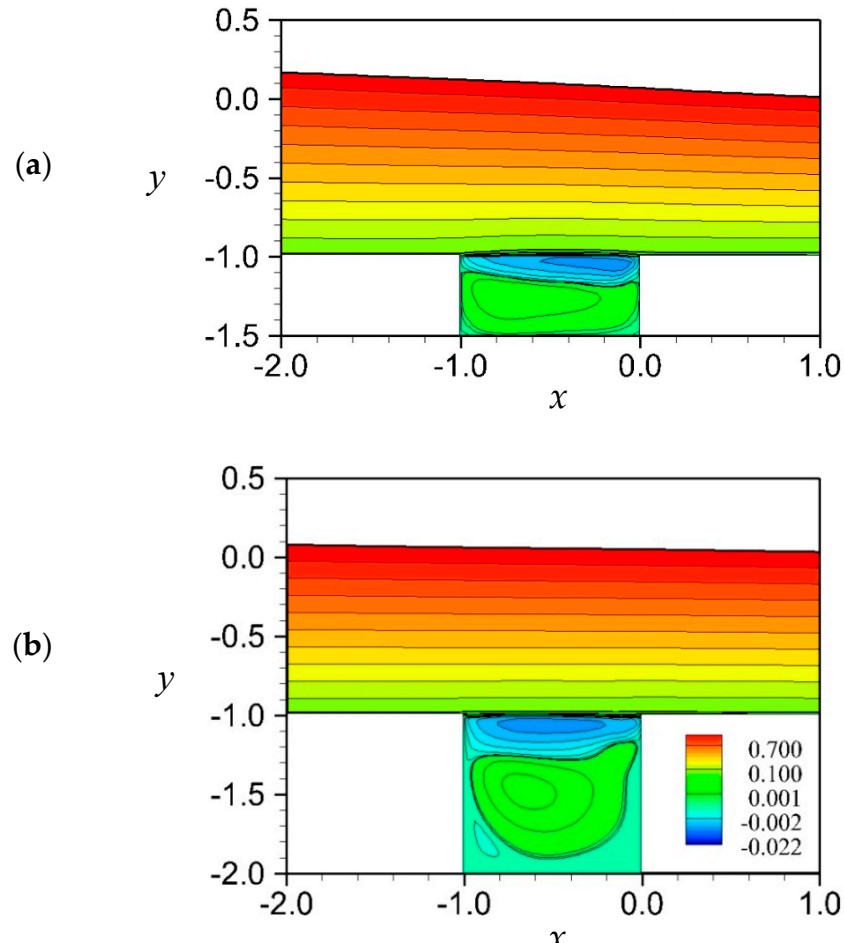

**Figure 16.** Streamline patterns in a cavity for *Re* = 5000 and *Fr* = 1.0 at *t* = 300 (quasi- steady phase) with cavity size (**a**) 1.0 × 0.5, and (**b**) 1.0 × 1.0.

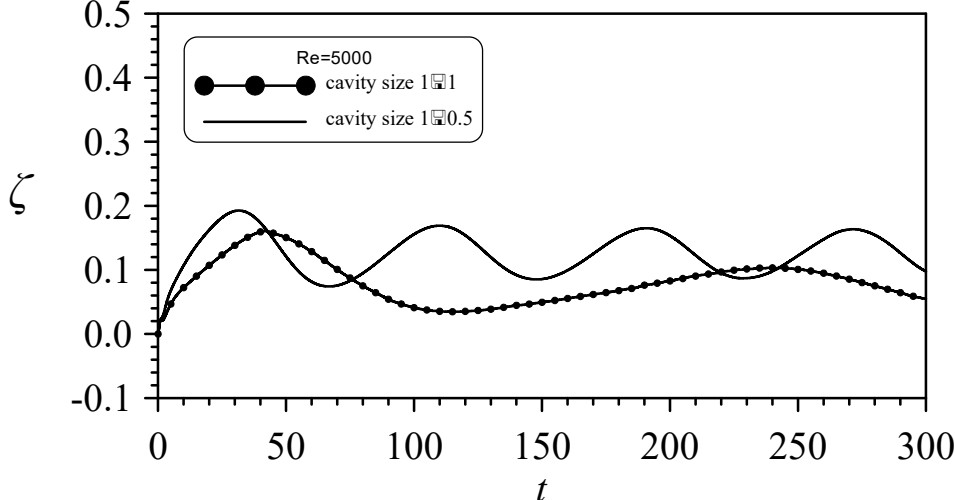

**Figure 17.** Time varying free-surface elevations for *Re* = 5000 at *Fr* = 1 with different cavity depth.

## 5. Conclusions

This numerical study presents the phenomena of free-surface deformation with generated upstream advancing solitons and downstream oscillating waves together with the vortical motion due

to a uniform flow passing over a bottom cavity. Under the effect of a negative forcing, the present model results are verified with the limiting cases of potential flow solutions of Zhang and Chwang [20]. In addition, the vortical-flow patterns in the cavity are compared with the results from other studies of pure lid-driven cavity flow. The breaking points and important findings are summarized as follows:

1.  This paper numerically explores the properties of a viscous free-surface flow over a cavity, which has never been investigated in the past for the vortex motions and the surface waves produced by a negative forcing. Numerical simulations under the consideration of a fixed cavity and various flow conditions ranging *Fr* from 0.5 to 1.1 at *Re* =500 and 5000 are carried out with results presented and discussed.

2.  Under the condition of *Fr* = 1.0, the vortical flows in the cavity for the cases of lower value of *Re* (e.g., *Re* = 500) are shown to be similar to the classical closed lid-driven cavity flow pattern, where a steady-state solution can be reached. For the case with a higher *Re* (e.g., *Re* = 5000), the flow motions, although can establish nearly to a quasi-steady state condition, are more complex and are very different from the patterns of *Re* = 500. Although the cavity flow can reach the steady state, the free surface will remain unsteady. This is a phenomenon worth exploring, one that not been explored by researchers. Therefore, we attempt to provide further possible phenomena in this area of research. As for the viscous influence on the wave development, at a lower *Re*, stronger advancing solitary waves are generated, possibly because of the increase in the thickness of the boundary layer on the solid walls and the shallower separation-streamline lid of the cavity. This problem is complicated because of the interaction between waves and vortices. Previous researchers have considered inviscid fluid for this problem. This will be very different from the real situation. Therefore, this paper hopes that research in this area can make further progress to consider the effect of viscosity.

3.  The wave properties of flows over a cavity is found to resemble those with flows over a hump. The forming of a movable but slightly protruding cavity lid as flows passing over a concavity becomes a forcing mechanism. With the values of *Fr* ranging from 0.5 to 1.1, a series of cases covering from subcritical to supercritical flow regimes are investigated. The results for the lower subcritical flow (e.g., *Fr* = 0.5) condition indicate that the water surface is disturbed with very weak undular motions. With an increase of *Fr* (e.g., up to lower supercritical flow conditions), it is noticed the wave height of the upstream advancing waves (solitons) increases. The time required for the development of each emerging solitary wave is also increased.

4.  In the literature, the variations in waves caused by flow over depressed terrain have not been widely discussed. When it is, the influence of viscosity is generally missing. Also, this phenomenon is difficult to produce in experiments. That is why the results for cavity flow and free surface were verified separately. We strove to prove the credibility of this model, although indirectly. Therefore, the motivation behind, and purpose of this paper, is to provide some information for investigators who are interested in this issue.

**Author Contributions:** The author (C.-H.C.) developed the numerical model, performed the computation, and drafted and polished the paper.

**Funding:** This research was funded by the Ministry of Science and Technology, Taiwan, Republic of China (Grant No. MOST 106-2221-E-275-003) and the APC for publishing this paper was also funded by the Ministry of Science and Technology, Taiwan, Republic of China (Grant No. MOST 106-2221-E-275-003).

**Acknowledgments:** This work was supported financially by the Ministry of Science and Technology, Taiwan, Republic of China (Serial No. MOST 106-2221-E-275-003). The author is very grateful to Professor Keh-Han Wang of the Department of Civil Engineering at the University of Houston for the improvement of many comments in the article. The authors would like to thank Enago (www.enago.tw) for the English language review.

**Conflicts of Interest:** The author declares no conflict of interest.

**Nomenclature**

| | |
|---|---|
| $\widetilde{A}, \widetilde{B}$ | convective coefficients |
| $b$ | shape of bottom object |
| $b_m$ | minimum tip of object |
| $Fr$ | Froude number |
| $g$ | gravitational acceleration |
| $g^{ij}, f^i$ | geometric coefficients |
| $H$ | non-dimensional still-water depth |
| $H^*$ | dimensional still-water depth |
| $(i, j)$ | grid-node indices |
| $IM$ | maximum grid index in $x$-direction |
| $J$ | Jacobian |
| $JM$ | maximum grid index in $y$-direction |
| $L$ | length of object |
| $\widetilde{n}$ | unit-normal vector |
| $Re$ | Reynolds number |
| $Re_{lid}$ | $Re$ for lid-driven cavity flow |
| $(U, V)$ | contra-variant fluid velocities |
| $U^*$ | dimensional inlet velocity |
| $u_f$ | tangential free-surface fluid particle velocity |
| $(x, y)$ | Cartesian coordinates |
| $\alpha$ | solitary-wave height |
| $\delta_\tau, \delta_\xi, \delta_\eta$ | finite-difference operators |
| $\Delta$ | variable increment |
| $\Delta n$ | normal distance between the wall and the adjacent node |
| $\nabla^2$ | Laplacian operator |
| $\zeta$ | free-surface elevation |
| $\nu$ | kinematic viscosity |
| $(\zeta, \eta)$ | Curvilinear coordinates |
| $v$ | time in transient curvilinear coordinate system |
| $\Psi$ | Stream function |
| $\Psi_1$ | stream function at the first grid node to the wall |
| $\omega$ | vorticity |

**Appendix A. Numerical Method for Free-Surface Calculation**

The free-surface elevation ($\zeta$) is calculated from the kinematic free-surface boundary condition (Equation (6)) whereas, the stream function ($\psi$) on the free surface is evaluated in accordance with the dynamic free-surface condition (Equation (7)).

We define the following operators with a dummy variable $f$,

$$\delta_\tau f = \frac{f_{i,JM}^{*(n+1)} - f_{i,JM}^n}{\Delta \tau} \tag{A1-a}$$

$$\delta_\xi f = \frac{f_{i+1,JM}^{*(n+1)} - f_{i-1,JM}^{*(n+1)}}{2\Delta \xi} \tag{A1-b}$$

$$\delta_\eta f = \frac{f_{i,JM}^{*(n+1)} - f_{i,JM-1}^{*(n+1)}}{\Delta \eta} \tag{A1-c}$$

where the superscript "*(n + 1)" represents the variables with temporary values used in each iteration at the (n + 1) new time level. After a series of calculation tests, a fully implicit second-order upwind scheme is utilized on the free-surface boundaries. In one of authors' early studies [31] on modeling flows over a bottom-mounted square cylinder, a mixed explicit–implicit scheme was used and was

found to be stable and accurate for the flow in the context of a positive-forcing problem. However, in this study, using a fully implicit second-order upwind scheme is shown to be a better approach in solving the problem of flows interacting with a negative forcing like body. Therefore, Equations (6) and (7) can be discretized and arranged as

$$\zeta_i^{n+1} = \zeta_i^n - \Delta\tau \left[ \frac{\delta_\xi \psi}{\delta_\xi x} \right] \tag{A2}$$

$$\psi_{i,JM}^{n+1} = \left[ -\frac{C_1}{2}\psi_{i+1,JM}^{*(n+1)} + \frac{C_1}{2}\psi_{i-1,JM}^{*(n+1)} + C_2\psi_{i,JM-1}^{*(n+1)} + C_3 + C_4 \right]/C_2 \tag{A3}$$

where

$$C_1 = Jg^{12} + \widetilde{A} = Jg^{12} + \Delta\tau \left[ -\delta_\tau(\frac{x_\eta}{J})\delta_\xi x - \delta_\tau(\frac{\zeta_\eta}{J})\delta_\xi\zeta \right]$$

$$C_2 = Jg^{22} + \widetilde{B} = Jg^{22} + \Delta\tau \left[ \delta_\tau(\frac{x_\xi}{J})\delta_\xi x + \delta_\tau(\frac{\zeta_\xi}{J})\delta_\xi\zeta \right]$$

$$C_3 = -\Delta\tau \left[ u^{*(n+1)}u_\xi + v^{*(n+1)}v_\xi + \delta_\xi\zeta \right]$$

$$C_4 = Jg^{12}\delta_\xi\psi + Jg^{22}\delta_\eta\psi$$

In coefficient $C_3$, $u_\xi$ and $v_\xi$ (generalized as $\varphi_\xi$ in expressions shown below) are modelled with a second-order upwind scheme as

$$\varphi_\xi = \begin{cases} (\varphi_{i-2,JM}^{*(n+1)} - 4\varphi_{i-1,JM}^{*(n+1)} + 3\varphi_{i,JM}^{*(n+1)})/2, \; if \; \varphi^{*(n+1)} > 0, \\ \\ (-\varphi_{i+2,JM}^{*(n+1)} + 4\varphi_{i+1,JM}^{*(n+1)} - 3\varphi_{i,JM}^{*(n+1)})/2, \; if \; \varphi^{*(n+1)} < 0. \end{cases} \tag{A4}$$

In addition, the finite difference method is utilized on the downstream boundary conditions, which are discretized backward in space using the first-order scheme and with the first-order temporal difference in this calculation.

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
