# Peer review of "Numerical Analyses of Wave Generation and Vortex Formation under the Action of Viscous Fluid Flows over a Depression"

_jmse, doi:10.3390/jmse7050141_

Reviewer 1 Report

Review of the Journal of Marine Science and Engineering manuscript entitled «Numerical analyses of wave generation and vortex formation under the action of viscous fluid flows over a depression» by Chih-Hua Chang

The paper considers the two-dimensional problem of the behavior of the free surface of a viscous fluid flowing over a submerged cavity in the form of an infinite rectangular channel. New interesting results were obtained on the study of various regimes in a wide range of Reynolds numbers. I believe that the work can be published in the Journal after considering the following minor remarks:

1. It seems to us incorrect to mention the study of the study of ocean waves in the context of this paper. Oceanic movements should be studied taking into account the rotation of the fluid plane (or sphere), where the impact of bottom reliefs is manifested in the form of generation of vortex structures with vertical axes.

2. I propose to draw the author’s attention to the articles:

E.A. Ryzhov, K.V. Koshel. «Steady and perturbed motion of a point vortex along a boundary with a circular cavity», Physics Letters A, 380, 896-902 (2016)

E.A. Ryzhov, K.V. Koshel, M.A. Sokolovskiy, X. Carton. «Interaction of an along-shore propagating vortex with a vortex enclosed in a circular bay», Physics of Fluids, 30, 016602 (2018)

where problems of vortex capture in an inviscid fluid are discussed by a quasi-circular cavity.

Reviewer 2 Report

The paper describes simulations of waves passing over an indented bottom. Results are validated by comparing with simulation results from the literature. A number of comments and questions:
= The numerical method makes use of a streamfunction-vorticity formulation, hence the method only works in two dimensions. Why not simply use the primitive variables? No rewriting of the equations including the free-surface conditions, and extendible to 3D.
= The grid size is selected at \Delta = 0.02, because (bottom of page 7) "a finer grid size of \Delta = 0.02 costs vast computational efforts". This is not an appropriate criterion to select a grid. Instead, accuracy should determine it. At least, a serious indication of discretization error should be provided. I think this point is essential since the first comparison with literature in Fig. 3b versus 3c, shows clearly different results. So, at least one of the simulations is inaccurate. The author must show that his results can be trusted sufficiently. The completely different first-order upwind results give me some doubts about this point.
= In section 3.2 details about the numerical algorithm are presented. It solves the equations implicitly at each time level. A reader would like to know how stable the involved iterations are. is there something to watch out for?
= Section 4.1.2 shows a comparison of results for a driven cavity. This is not really impressive as the reference results come from a period with much less computer power. Grid refinement info would be welcome. 
= Then three pages of colorful pictures follow, but what do we learn from them? These pictures look very much alike. What are the essential details of the flow which are important for the final test case?
= Section 4.4 shows the final test case of waves passing over a rectangular indentation at several Froude numbers. Close to the critical value, the "flows create dramatically different wave phenomena" (line 411). At such moments, I desperately need some information about the numerical accuracy, before I dare to interpret the results. 
= In summary, I have no idea what the results are worth. Are they reliable enough to be judged physically? Without a sufficient demonstration of the numerical uncertainty, the paper cannot be accepted for publication.

Reviewer 3 Report

Attached

Author Response

Round  2

Reviewer 2 Report

The paper has certainly improved a bit. But I found the author's reaction to my earlier comments more interesting. Please add some of that information into the paper. In particular the reaction to my comments 3, 4 and 5, with figures R2-1 (in a better format) and R2-2, should be added tot the paper. Also some sentences from reaction 6 and 7 could be added. This will bring numerical reliability and confidence to the results, which is needed before the results may be interpreted physically..    

Author Response

Please see the attached PDF file. Thanks.

Reviewer 3 Report

attached

Author Response

(The authors gave the same response as above.)
